# Respiration modulates oscillatory neural network activity at rest

**Daniel S. Kluger** [1,2]*, **Joachim Gross**[1,2,3]

**1** Institute for Biomagnetism and Biosignal Analysis, University of Muenster, Muenster, Germany, **2** Otto Creutzfeldt Center for Cognitive and Behavioral Neuroscience, University of Muenster, Muenster, Germany, **3** Centre for Cognitive Neuroimaging, Institute of Neuroscience and Psychology, University of Glasgow, Glasgow, United Kingdom

\* daniel.kluger@wwu.de

## Abstract

Despite recent advances in understanding how respiration affects neural signalling to influence perception, cognition, and behaviour, it is yet unclear to what extent breathing modulates brain oscillations at rest. We acquired respiration and resting state magnetoencephalography (MEG) data from human participants to investigate if, where, and how respiration cyclically modulates oscillatory amplitudes (2 to 150 Hz). Using measures of phase–amplitude coupling, we show respiration-modulated brain oscillations (RMBOs) across all major frequency bands. Sources of these modulations spanned a widespread network of cortical and subcortical brain areas with distinct spectrotemporal modulation profiles. Globally, delta and gamma band modulations varied with distance to the head centre, with stronger modulations at distal (versus central) cortical sites. Overall, we provide the first comprehensive mapping of RMBOs across the entire brain, highlighting respiration–brain coupling as a fundamental mechanism to shape neural processing within canonical resting state and respiratory control networks (RCNs).

## Introduction

We all breathe. Human respiration at rest comes naturally and comprises active (but automatic) inspiration and passive expiration [1]. The rhythmicity of each breath is initiated and coordinated by coupled oscillators periodically driving respiration, most prominently the preBötzinger complex located in the medulla [2]. This microcircuit typically controls respiration autonomously, making the act of breathing seem effortless. Importantly, however, respiration is also under top-down control, as evident from adaptive breathing during, e.g., speaking, laughing, and crying [3]. Hence, there is a bidirectional interplay of the cortex and rhythmic pattern generators of respiration: Efferent respiratory signals from the preBötzinger complex project to suprapontine nuclei (via locus coeruleus and olfactory nuclei [4]) as well as to the central medial thalamus, which is directly connected to limbic and sensorimotor cortical areas [5]. In turn, cortical areas evoke changes in the primary respiratory network, e.g., to initiate specific motor acts (e.g., swallowing or singing) or transitions between brain states (e.g., during panic attacks).

As neural oscillations have been established as sensitive markers of brain states in general [6], the question arises to what extent rhythmic brain activity is modulated by the rhythmic act

**Data Availability Statement:** The anonymised data supporting the findings of this study are openly available from the Open Science Framework (https://osf.io/6zbdu/).

**Funding:** This work was supported by the Interdisciplinary Center for Clinical Research (IZKF)

of the medical faculty of Muenster, Grant Number Gro3/001/19 (JG) and the German Research Foundation, Grant Number GR 2024/5-1 (JG). The funders had no role in study design, data collection and analysis, decision to publish, or preparation of the manuscript.

**Competing interests:** The authors have declared that no competing interests exist.

**Abbreviations:** ACC, anterior cingulate cortex; aIPS, anterior intraparietal sulcus; DAN, dorsal attention network; DMN, default mode network; FEF, frontal eye field; fMRI, functional magnetic resonance imaging; iEEG, intracranial EEG; IPS, intraparietal sulcus; ITG, inferior temporal gyrus; LCMV, linearly constrained minimum variance; LMEM, linear mixed effect model; MEG, magnetoencephalography; MI, modulation index; MNI, Montreal Neurological Institute; mPFC, medial prefrontal cortex; MRI, magnetic resonance imaging; MTG, medial temporal gyrus; NMF, nonnegative matrix factorization; OB, olfactory bulb; OFC, orbitofrontal cortex; PCC, posterior cingulate cortex; PTA, phase-triggered average; RCN, respiratory control network; RMBO, respiration-modulated brain oscillation; SMA, supplementary motor area; SN, salience network; STG, superior temporal gyrus; TPJ, temporoparietal junction; VAL, ventral anterior lateral.

of breathing. Indeed, studies of respiration–brain coupling have recently attracted increased attention, reporting a range of cognitive and motor processes to be influenced by respiration phase. Human participants were found to spontaneously inhale at onsets of cognitive tasks [7] and respiration phase modulated neural responses in sensory [8] and face processing [9] tasks as well as during oculomotor control [10] and isometric contraction [11]. Parallel to this body of work, animal studies have conclusively shown respiration to entrain brain oscillations not only in olfactory regions [12], but also in rodent whisker barrel cortex [13] and even hippocampus [14]. In other words, brain rhythms previously attributed to cognitive processes such as memory were demonstrated to at least in part reflect processes closely linked to respiration [15].

Despite significant advances in the animal literature, these links are still critically understudied in humans. Notable exceptions include intracranial EEG (iEEG) work in epilepsy patients corroborating that oscillations at various frequencies can be locked to the respiration cycle even in nonolfactory brain regions [9]. Moreover, 2 noninvasive studies recently linked respiration phase to changes in task-related oscillatory activity [7]. Overall, both animal and human studies all lead to 3 fundamental questions that recognise respiration as a vital, continuous rhythm persisting during all tasks and activities as well as at rest: (i) to what extent does breathing modulate rhythmic brain oscillations at rest; (ii) where are these modulatory effects localised in the brain; and (iii) how does modulation unfold over the course of the respiration cycle. Therefore, what is needed is a comprehensive account integrating recent findings of respiration–brain coupling against the anatomical backdrop of canonical resting state and respiratory control networks (RCNs). A variety of neural networks have extensively been described to organise the brain's intrinsic or ongoing activity, among which the default mode network (DMN), the dorsal attention network (DAN), and the salience network (SN) have received particular attention [16]. Previous studies have demonstrated intriguing anticorrelated dynamics of activity between these large-scale networks (i.e., increases in one network lead to decreases in another [17]). Such fluctuating relationships between cortical networks could conceivably be modulated by changes in bodily states such as respiration. The full picture is complemented by 2 distinct pathways responsible for the feedforward generation of the respiratory rhythm and the neural processing of respiration-related signals, respectively: In addition to pattern generators like the preBötzinger complex in the medulla, deeper sites known to be involved in respiratory control comprise further subregions within the brain stem [18] and cerebellar nuclei [19]. On the other hand, nasal respiration evokes feedback signalling in response to mechanical, thermal, or odour stimulation within olfactory areas in the forebrain, most prominently the olfactory bulb (OB) and piriform cortex [12]. These (orbito-)frontal feedback signals are a central contributor for respiration–brain coupling, as animal studies have demonstrated that respiratory rhythms (i.e., air-driven mechanoreceptor signals within the OB) are translated into neural oscillations and propagated to upstream areas [13]. Interestingly, the RCN also includes directly connected cortical sites like primary and supplementary motor areas (SMAs) [20] and even shows anatomical overlap with resting state networks, namely within medial prefrontal cortex (mPFC) [21], insula [22], and anterior cingulate cortex (ACC) [23]. We thus aimed to investigate respiration-related modulations of oscillatory brain activity and its spectrotemporal characteristics at rest, relating their anatomical sources to canonical networks of both resting state activity and respiratory control.

To this end, we simultaneously recorded spontaneous respiration and eyes-open resting state magnetoencephalography (MEG) data from healthy human participants. Using the modulation index (MI) as a measure of cross-frequency phase–amplitude coupling [24], we first assessed respiration-induced modulation of brain oscillations globally across the entire brain. We then extracted single-voxel time series to localise the anatomical sources of these global

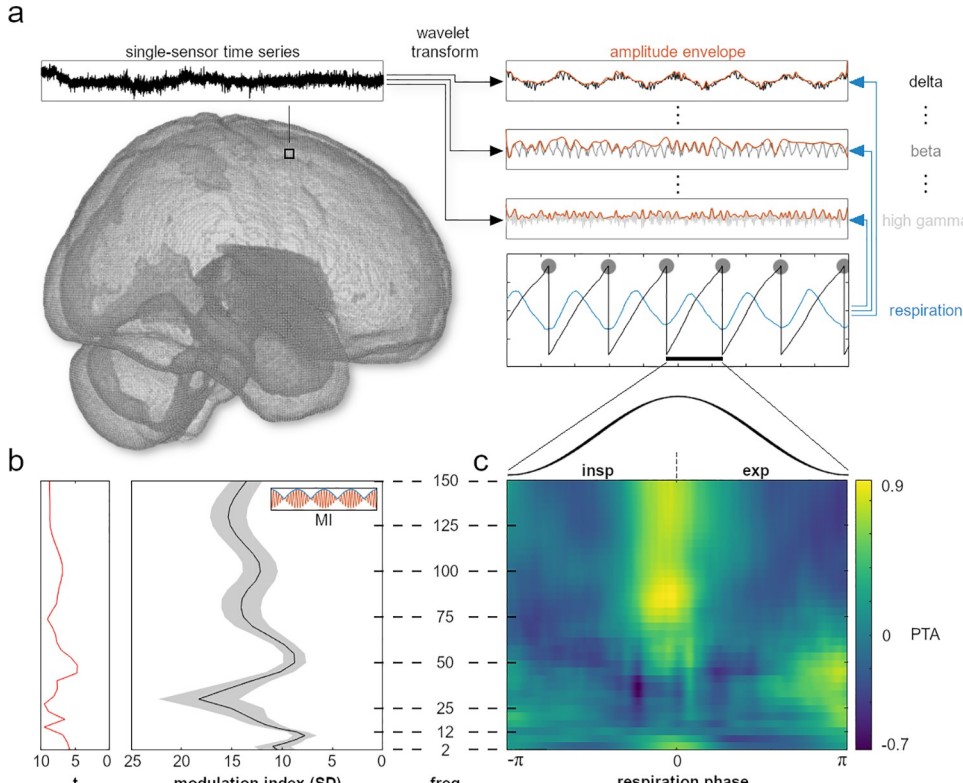

**Fig 1. Respiration-induced modulation of global field power. (A)** Exemplary schematic of our analysis approach showing the wavelet transform of time series data from each voxel. Global field power was computed on the time course of all 268 channels for further analyses of the MI and PTA. MI quantifies to what extent the amplitude envelopes of frequency-specific brain oscillations (top right, red) were modulated by respiration (centre right, blue). This way, we computed modulation indices for each sensor, frequency, and participant before localising voxelwise time series in source space (see Fig 2). **(B)** Mean normalised MI (± SEM) over the entire frequency spectrum (right) and corresponding t-values from the cluster permutation test (left). Random shifts of respiration phase were employed to correct for low-frequency bias and to express MI in units of SD of a surrogate distribution (leading to normalised MI; see Materials and methods). **(C)** Mean PTAs across the respiratory cycle over the entire frequency spectrum. PTAs were computed by averaging frequency-specific amplitude envelopes (panel A) time locked to peak inhalation. Note that PTAs were standardised for illustration purposes. Therefore, they show relative changes over the respiration cycle within each frequency band and do not directly correspond to absolute MI amplitudes from panel B. Also note that SNR decreases towards the edges of the panel (i.e., approaching ± π) due to increased variation of underlying single respiratory cycles that were used for phase-locked analysis. Underlying data are provided in the folder "Fig 1" on the OSF directory. MI, modulation index; PTA, phase-triggered average; SD, standard deviation; SNR, signal-to-noise ratio.

modulation effects using beamforming (Fig 1A). We employed nonnegative matrix factorisation (NMF) for dimensionality reduction, effectively yielding a spatially constrained network of cortical and subcortical sources of respiration phase–dependent changes in rhythmic brain activity. Finally, we identified distinct spectrotemporal profiles of network components, highlighting an intriguing organisational pattern behind respiration-induced modulation of neural oscillations across the brain.

## Results

### Respiration phase modulates global field power

To assess the fundamental question of whether respiration modulates oscillatory brain activity at rest, we first computed the MI in sensor space (using all 268 channels) for whole-brain

global field power ranging from 2 to 150 Hz. This analysis quantifies to what extent the amplitude of global brain oscillations is modulated by the phase of respiration. Our cluster permutation analysis revealed significant respiration-locked modulation of global field power indicated by the high normalised MI across the entire frequency spectrum (all $p < 0.001$, cluster corrected at $\alpha = 0.05$; see Fig 1B). Local peaks with strongest modulation occurred at about 2, 30, 75, and 130 Hz (with strongest absolute modulation effects in the beta band), indicating differential modulation of specific brain oscillations (see S1 Fig for range and distribution of subject-level MI spectra). Next, we computed the phase-triggered average (PTA) to characterise these global modulation effects over a respiratory cycle. PTA is computed as the average of oscillatory amplitude across windows centred on all time points of peak inhalation. We found respiration phase to differentially modulate oscillations of various frequencies with distinct time courses. In particular, whereas most frequency bands showed strongest modulation effects around the inspiration peak, beta oscillations were visibly coupled to a different phase of respiration around inspiration onset (Fig 1C; see S2 Fig for subject-level PTA spectrograms).

This first analysis therefore revealed that the amplitude of global oscillatory brain activity was significantly modulated by respiration in a broad frequency range from 2 to 150 Hz with a temporal modulation that differs across frequencies. To gain a deeper understanding of how respiration modulates rhythmic activity across the brain, 2 questions immediately ensued, namely to localise the anatomical sources of such modulation effects and to explore their spectrotemporal profiles in more detail.

## Modulatory effects of respiration phase can be traced to cortical and subcortical networks

To identify the anatomical sources of these global modulations, we quantified how strongly respiration modulated the amplitude of brain oscillations within each voxel in the brain of each participant at each frequency between 2 and 150 Hz by computing the MI. Next, we used sparse NMF to reduce the dimensionality of the three-dimensional data set (participants × voxels × frequency; see Materials and methods). This resulted in an optimal low-dimensional representation consisting of 18 components. Each component reflected respiration-modulated brain oscillations (RMBOs) across the frequency spectrum, quantified as NMF weights for each participant, voxel, and frequency. For spatial specificity of NMF components, each component's spatial map was thresholded at the 99th percentile, yielding the $n = 202$ voxels with the strongest modulation. For all 18 components, we show the spatial location of the network on an inflated brain as well as the full MI spectrum with shading corresponding to frequency bands of significant modulation (all $p < 0.002$, cluster corrected across frequencies and components at $\alpha = 0.05$; see Fig 2). For individual spatial maps for all 18 components, see S3 and S4 Figs. Phase-dependent modulations of all 18 components are shown in S5 Fig. Together, this provides a comprehensive spatiotemporal spectral account of respiration-modulated networks in the resting brain.

Fig 2A shows the network's cortical sources to be localised along the midline (ACC, SMA, posterior cingulate cortex (PCC), cuneus, and lingual sulcus) as well as in lateralised frontal (frontal eye field (FEF) and insula), temporal, and parietal cortices (angular gyrus and intraparietal sulcus (IPS)). The network's deeper, subcortical sites included several lateralised (crus 1, lobules 7b/8) and midline (vermis 9/10) subsections within the cerebellum, left parahippocampal cortex, and medial sources in the orbitofrontal cortex (OFC; extending onto the gyrus rectus) and brain stem (Fig 2B). In order to quantify modulation effects between components, we compared each component's MI at a given frequency with the average MI across all other

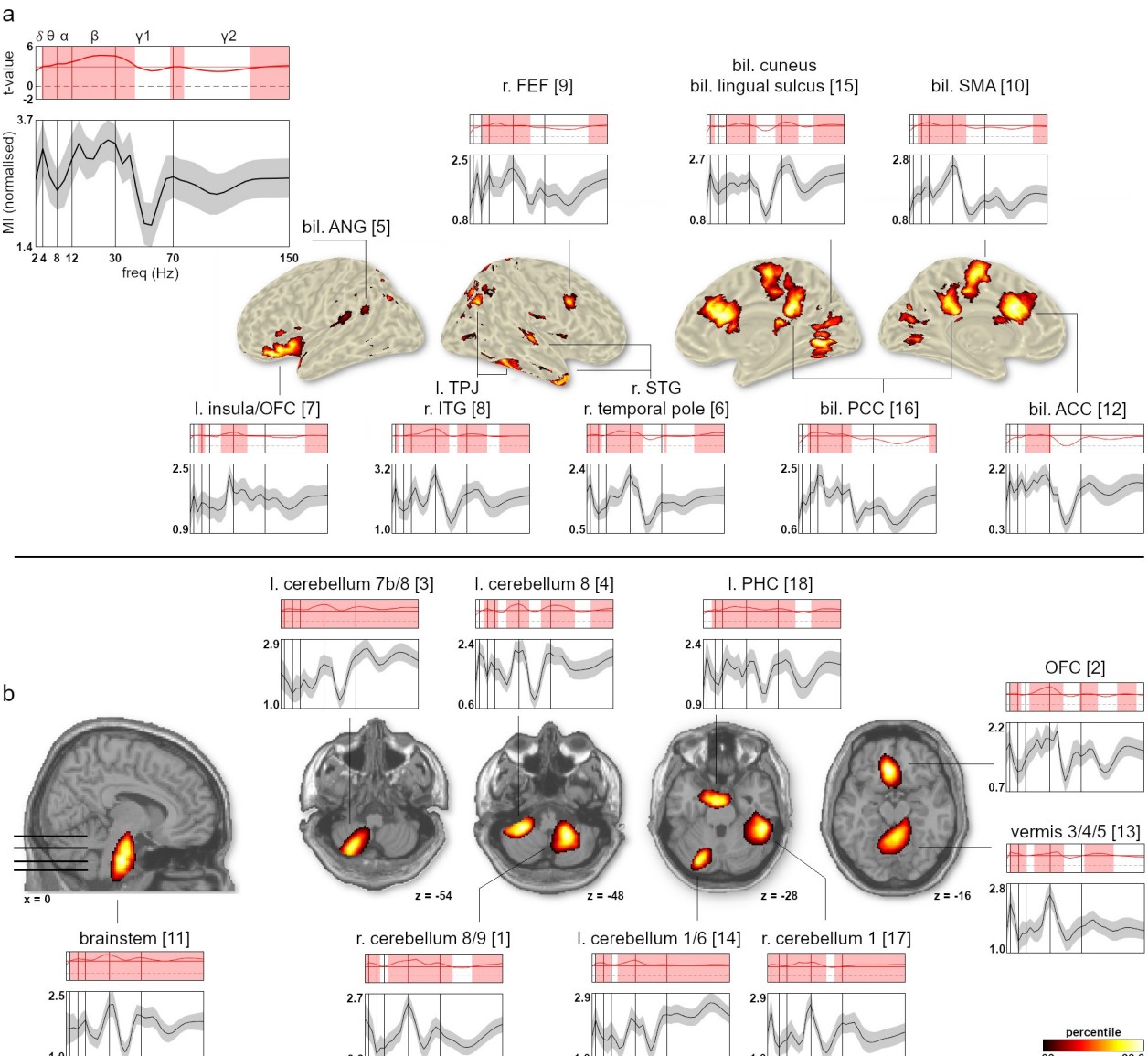

**Fig 2. Anatomical locations and spectral modulation profiles of NMF components whose neural oscillations were significantly modulated by respiration. (A)** Cortical components plotted on an inflated brain surface. Bottom graphs illustrate each component's normalised average MI course across frequencies (2 to 150 Hz). Upper graphs show component-specific t-value spectra from the cluster permutation test (with significant cluster-corrected frequencies shaded). Horizontal red line marks the significance threshold of each component, and vertical lines mark borders between frequency bands (delta to high gamma). Spatial maps were thresholded at the 99th percentile across all 20,173 voxels of each component, yielding the $n = 202$ voxels with the strongest modulation. Colour bar indicates corresponding p-values. **(B)** Subcortical components plotted on transverse and sagittal slices of the MNI brain. Same format as A. Underlying data are provided in the folder "Fig 2" on the OSF directory. ACC, anterior cingulate cortex; FEF, frontal eye field; MI, modulation index; MNI, Montreal Neurological Institute; NMF, nonnegative matrix factorisation; OFC, orbitofrontal cortex; PCC, posterior cingulate cortex; PHC, parahippocampal cortex; SMA, supplementary motor area; STG, superior temporal gyrus; TPJ, temporoparietal junction.

components. Components #3 and #14 (both located within the left cerebellum) showed above average modulation in the high gamma band, whereas components #6 (r. STG/r. temporal pole) and #11 (brain stem) were more strongly modulated in the delta and alpha band, respectively. Component #10 (bil. SMA) showed above average modulation in the beta and low gamma range. Finally, component #12 (bil. ACC) was less strongly modulated at low gamma frequencies than the grand average across components (see S6 Fig).

These results provide several important insights. First, respiration significantly modulates oscillatory brain activity within a specific, but widely distributed cortical and subcortical brain network. Second, across these areas, significant RMBOs can be found across almost the entire frequency range from 2 to 150 Hz. Third, the temporal modulation pattern of RMBOs is by no means uniform across frequencies and brain areas.

## Distinct spectrotemporal profiles of RMBO sites

Having localised the anatomical network underlying RMBOs, we next attempted to map distinct modulation patterns to anatomical subnetworks, with similarly modulated sites being grouped together. To this end, we employed hierarchical clustering of all 18 network components based on their MI across the frequency spectrum (as shown in Fig 2). This data-driven approach yielded a total of 7 clusters comprising between 1 and 5 components (Fig 3; see S7 Fig and Materials and methods for details): Cluster A consisted of a single component within the left insular cortex/OFC and showed significant modulation across all frequency bands except the delta band. Cluster B showed a clear cortical organisation along the midline, comprising bilateral PCC and SMA components with significant modulation from theta up to high gamma oscillations. Similarly, cluster C comprised 2 components within bilateral ACC and right FEF with significant modulation from alpha up to high gamma oscillations. Cluster D was formed by a total of 6 components spanning inferior, medial, and superior temporal gyrus (ITG, MTG, and STG), parietal cortices (anterior intraparietal sulcus (aIPS)/temporoparietal junction (TPJ), and angular gyrus) as well as deep cerebellar areas showing RMBOs. Due to its widespread topography, at least 1 cluster component showed significant modulation across the entire frequency spectrum. Cluster E again consisted of a single component (spanning bilateral (pre-)cuneus/lingual sulcus) with significant modulation in the theta, beta, and both gamma bands. Finally, 2 clusters were formed exclusively by deep sources: Cluster F comprised 2 components within the left cerebellum where oscillations across the whole frequency spectrum were significantly modulated by respiration. Cluster G consisted of 4 components within the left parahippocampal cortex, brain stem, cerebellum, and gyrus rectus/medial OFC and showed significant modulation from theta up to the high gamma band.

In order to investigate how MI spectra varied with anatomical location, we conducted a linear mixed effect model (LMEM) analysis modelling oscillatory modulations as a function of components' distance to the head centre (considering x, y, and z planes). This analysis revealed that the fixed effect of distance to the head centre significantly influenced modulations within the delta ($t(502) = 3.55$, $p < 0.001$) as well as the low ($t(502) = 2.49$, $p = 0.013$) and high gamma bands ($t(502) = 3.85$, $p < 0.001$), with stronger modulations for more distal (compared to central) components. Further, frequency-specific analyses were conducted to characterise to what extent this overall distance effect was driven by sagittal, frontal, and transversal location, respectively (see S1 Text and S1 Table for details).

Intriguingly, not only were different frequency bands modulated within a network of cortical and subcortical sites, but the time courses of these modulatory effects were equally frequency specific. Polar plots in Fig 3 show the temporal modulation of RMBOs across the respiratory cycle for each cluster. Respiration phase was differentially coupled with amplitudes of low-frequency oscillations (such as delta and theta) compared to high-frequency oscillations (e.g., within the gamma band). Low frequencies consistently showed higher amplitudes during the beginning and end of a respiration cycle (with lowest amplitudes around peak inspiration), whereas the pattern appeared reversed for higher frequencies (see Fig 3). While specific spatio-temporal interactions of respiration–brain coupling exceeded the conceptual scope of this

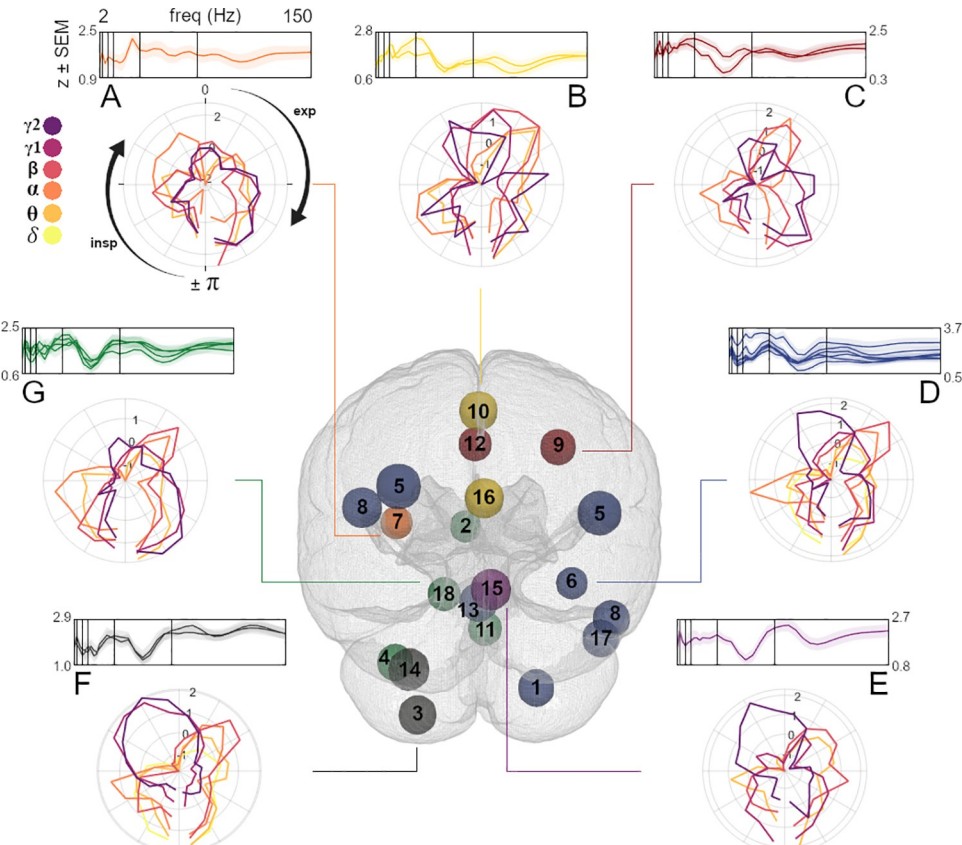

**Fig 3. Time-frequency characteristics and anatomical distribution of component clusters.** Hind view of the glass brain illustrates the spatial distribution of component clusters A to G (numeration according to Fig 2; see S8 Fig for top and side views). Spheres mark peak locations of components and are coloured according to cluster affiliation. Top curve plots depict z-transformed modulation indices of individual components within the cluster (group-level mean ± SEM, identical to Fig 2) across all frequencies to visualise within-cluster similarities. Vertical bars mark borders between frequency bands (see Fig 2 and main text). Polar plots show cluster-average RMBOs (colour coded for frequency; see top left) as a function of respiration phase (where zero corresponds to the peak of the respiration signal). For clarity, polar plots are restricted to frequencies that were significantly modulated in at least 1 component of the respective cluster. Underlying data are provided in the folder 'Fig 3' on the OSF directory. exp, expiration; insp, inspiration; RMBO, respiration-modulated brain oscillation.

study, our findings are the first to suggest such spatiospectral gradients and thus warrant detailed examination in future work.

## RMBOs within nodes of resting state and RCNs

Extending the distinction of deep versus more superficial components, cortical RMBOs were predominantly found in brain areas that have previously been established as nodes within the DMN (PCC, angular gyrus, and precuneus), DAN (FEF and aIPS), or saliency network (SN: insula and ACC; see [15]). Moreover, all deep and cerebellar modulation sites corresponded to a mostly subcortical network of brain areas controlling respiratory function, including bilateral cerebellum, gyrus rectus/OFC, brain stem, and SMA (Fig 4C). Finally, as a potential link for future studies, Fig 4D appears to suggest that, although RMBOs of different frequencies had distinct temporal modulation profiles in general, there could also be certain sequential modulation patterns across clusters within a particular frequency. For example, while significant modulation of beta oscillations showed a general peak around expiration onset (distinct from

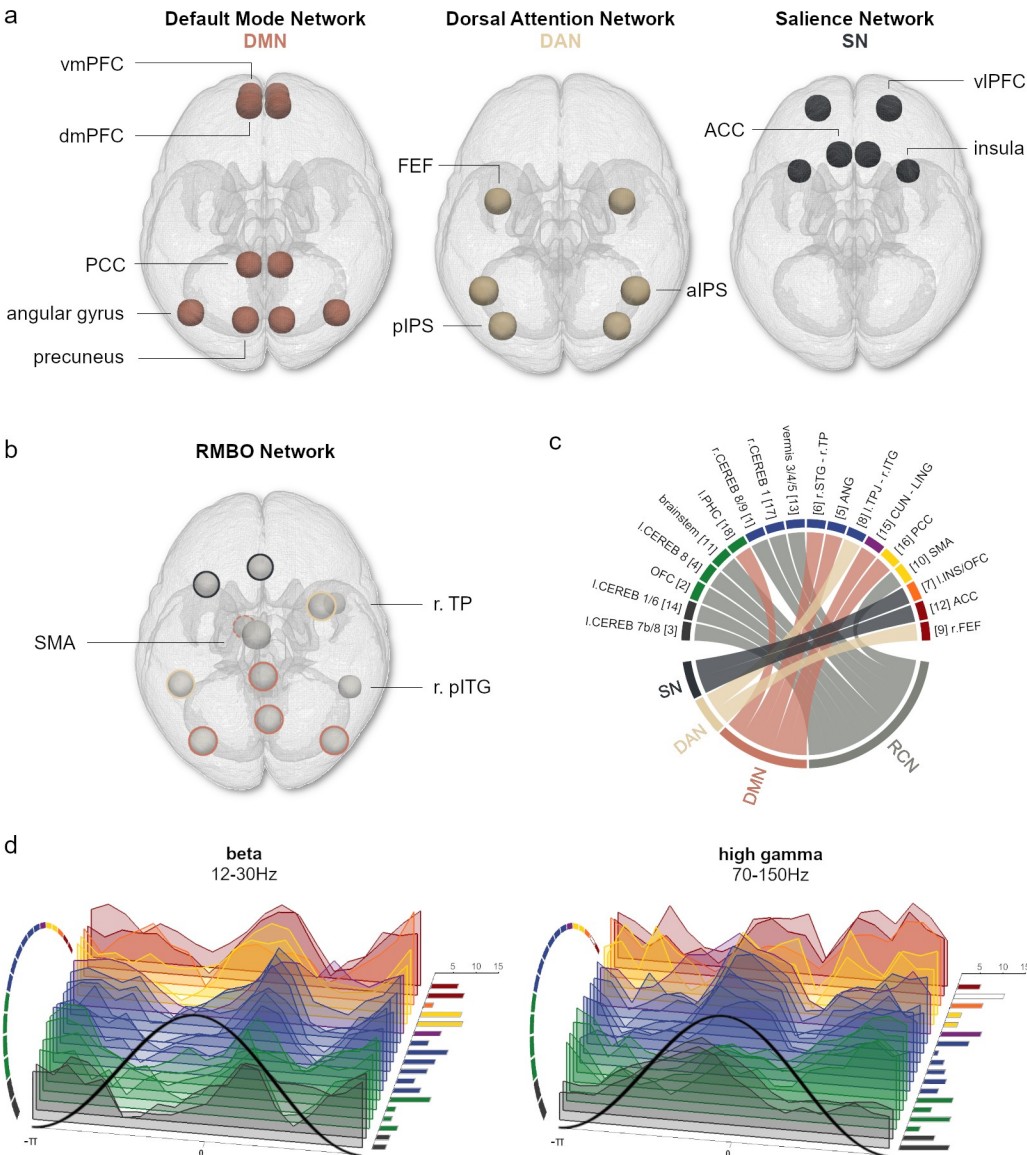

**Fig 4. Mapping clusters of NMF components to canonical neural networks. (A)** Top view stylised illustrations of neural nodes composing the DMN, DAN, and SN as described in the literature. **(B)** Cortical brain areas showing significant RMBOs (as in Fig 3) are colour coded according to their correspondence to the resting state networks shown in A. As the MTG has increasingly been included in the DMN but was not part of its original formulation, NMF components located within the MTG are marked with a dashed line. **(C)** Direct mapping of all 16 clustered NMF components to the resting state neural networks (see a) and the RCN gained from the literature. Colour code for clusters A to G taken from Fig 3. **(D)** Waterfall plots show z-transformed amplitude modulation phase locked to the respiration cycle exemplified for beta (left) and high gamma oscillations (right). Clusters of NMF components are shown in the same order as in c. Right panel bar graphs show the number of participants whose modulation within the respective component was strongest for the depicted frequency band (versus all other frequency bands). Coloured bars and circular segments mark NMF components for which the respective frequency band was significantly modulated by respiration phase. See S9 Fig for waterfall plots of the remaining frequency bands. Underlying data are provided in the folder "Fig 4" on the OSF directory. aIPS, anterior intraparietal sulcus; DAN, dorsal attention network; DMN, default mode network; dmPFC, dorsomedial prefrontal cortex; FEF, frontal eye field; MTG, medial temporal gyrus; NMF, nonnegative matrix factorisation; PCC, posterior cingulate cortex; pIPS, posterior intraparietal sulcus; pITG, posterior inferior temporal gyrus; RCN, respiratory control network; RMBO, respiration-modulated brain oscillation; SMA, supplementary motor area; SN, salience network; vlPFC, ventrolateral prefrontal cortex; vmPFC, ventromedial prefrontal cortex; TP, temporoparietal.

e.g., high gamma modulation), this peak appeared to occur earlier and less pronounced in cluster B (PCC and SMA) than in cluster D (cerebellum and temporoparietal cortex). Future work could aim to verify such latency effects between nodes of the RMBO network and their potential functional significance. At present, our results provide a unique perspective on the general link between respiration phase and changes in oscillatory activity, showing how the sources of these modulatory effects correspond to nodes of canonical networks in control of resting state activity and respiratory function.

## Discussion

Using noninvasive MEG recordings of human participants at rest, we performed the first spatially and spectrally comprehensive analysis of brain activity that is modulated by respiration. We identified RMBOs across the entire spectrum between 2 and 150 Hz within a widespread network of cortical and subcortical brain areas. The voxel-based analysis employed adaptive beamforming for source localisation. Adaptive beamforming optimally combines MEG recordings from all sensors to estimate the time series of neural activation at a given voxel while maximally suppressing interferences from other voxels. Although spatial resolution decreases with distance from sensors, it is generally suitable for cortical and subcortical areas. Intriguingly, instead of a uniform modulation pattern across brain areas and frequencies, our analysis revealed respiratory modulation signatures that differed between brain areas in frequency and the temporal modulation profile. Our results demonstrate that respiration significantly modulates oscillatory brain activity in a manner that is precisely orchestrated across frequency bands and networks of resting state activity and respiratory control. In what follows, we will integrate our novel results with the existing animal and human literature, characterise the functionality of neural oscillations within distinct networks, and attempt to provide an overview of potential multilevel mechanisms behind RMBOs.

### Subcortical and cortical sites of respiration–brain coupling

Gamma oscillations within the OB were the first to be described in detail [25] and reflect local computations within the OB [26]. In a next step, slower (e.g., beta band) oscillations are thought to organise such local activity across brain areas [27] and appear to be the most coherent within the OB [28]. Similarly, even slower theta oscillations play a crucial role in the temporal organisation of neural activity within the hippocampal network and, consequently, its coordination with the mPFC [29]. Our findings substantially advance these notions by showing that respiration phase modulates both low and high oscillatory frequencies within a spatial array comprising OB/OFC, brain stem, and cerebellum. As described earlier, the preBötzinger complex is widely regarded as the main pattern generator of respiratory rhythms within the brain stem [4], where ascending catecholaminergic neurons receive projections from the cerebellar vermis [30]. In addition to brain stem projections, the vermis regulates autonomic responses including cardiovascular tone and respiration through connections to the spinal cord and hypothalamus [31]. These cerebellothalamic pathways affect cortical gamma activity, in that cerebellar projections to the "motor" ventral anterior lateral (VAL) nucleus of the thalamus, the (higher order) posterior thalamic nucleus (VP), and intralaminar nuclei are used to coordinate and synchronise gamma oscillations within sensorimotor areas [32] and across the neocortex [33].

The cerebellum itself projects to motor and nonmotor cortical areas, including prefrontal and posterior parietal cortex [34]. In turn, it receives input from a wide range of higher-order, nonmotor areas within the extrastriate cortex, posterior parietal cortex, cingulate cortex, and the parahippocampal gyrus, which is monosynaptically connected to the OB [35]. As the first

olfactory relay station in the brain, the OB receives direct projections from receptors within the nasal cavity [36]. These feedback signals not only encode odour information in olfactory receptor cells [37], but also mechanical stimulation of mechanoreceptors triggered by nasal airflow [38]. As outlined above, subsequent neural activity patterns are then transmitted to upstream sites including OFC, hippocampus, and insula, which we fittingly found to be part of the RMBO network.

In sum, our findings integrate and extend a variety of individual results in 2 ways: First, cortical nodes within the RMBO network precisely reflect bidirectional projection areas of the deep and subcortical nodes (OB, brain stem, and cerebellum) via medullar and thalamic pathways. Second, the cortical nodes markedly resemble "sensorimotor distributions" shown in multiple functional magnetic resonance imaging (fMRI) studies of respiratory control [39], raising the question as to how different cortical areas—motor areas, ACC, and insular cortex —are involved in the act of breathing. As both premotor and supplementary motor cortices contain representations of respiratory muscles [40], they have long been implicated in respiratory control. Similarly, ACC has been identified in studies of air hunger [23] and $CO_2$-stimulated breathing. Finally, insular cortex activation is a consistent feature of many neuroimaging studies of dyspnoea [22]. The close mapping of frontal, cingulate, and parietal areas to canonical resting state networks (see Fig 4) suggests a general involvement of respiration in human brain processing irrespective of particular task demands. In this context, it is noteworthy that nodes of resting state networks exhibit amplitude correlations predominantly in the beta frequency band [41]. In our data, this frequency band shows strongest global modulation by respiration (Fig 1B) and features prominently in the coupling of specific resting state networks to respiration (Figs 3 and 4), suggesting that these amplitude correlations within resting state networks are at least partially related to respiration.

## Active sensing, respiration, and behaviour

The widespread extent and spectral diversity of the RMBO network critically corroborate previous suggestions of respiration as an overarching "clock" mechanism organising neural excitability throughout the brain [13]. Excitability adapts neural responses to current behavioural demands, which is why respiratory adaptation to such demands in animals [42] and humans [43] have accordingly been interpreted as functional body–brain coupling. Indeed, animals as well as humans appear to actively align their breathing to time points of particular behavioural relevance for the sake of efficiency through optimised neural processing. Consequently, human respiration has fittingly been cast as active sensing [44], adopting key premises from predictive brain processing accounts [45] to explain how respiration synchronises time frames of increased cortical excitability with the sampling of sensory information. Our results provide first insights into how established mechanisms like cross-frequency phase–amplitude coupling (in this case, coupling peripheral to neural rhythms) are implemented on a global scale to translate respiratory rhythms into neural oscillations of various frequencies and how the resulting anatomical pattern of RMBOs reflects spectral specificity.

## Potential mechanisms behind RMBOs

Cross-frequency coupling is widely regarded as the core mechanism of translating slow rhythms into faster oscillations and has conclusively been shown to be driven by respiratory rhythms within the OB in mice [13]: During nasal inspiration, air enters the respiratory tract and triggers mechanoreceptors connected to the OB, thereby initiating infraslow neural oscillations closely following the respiratory rhythm (phase–phase coupling). The phase of these slow oscillations then drives the amplitude of faster oscillations (phase–amplitude coupling)

and propagates to upstream areas both within and beyond the olfactory system [46]. With reference to the concept of active sensing introduced above, we argue that a similar case can be made for the cerebellum: There is broad consensus that the cerebellum is involved in computations attributed to internal forward models, predominantly in motor control [47]. These forward models are just as crucial for perception and cognition as they are for motor performance, leading to the suggestion that cerebellar processing may help to align and adaptively modify cognitive representations for skilled and error-free cognitive performance [34]. The prominent role of cerebellar sources in our present findings suggests that respiration modulates these functional connections by means of cross-frequency coupling, linking respiration to motor and cognitive function.

Strong support for this hypothesis comes from a recent study [48] showing that the cortical readiness potential, originating within premotor areas, fluctuates with respiration. Notably, the authors suggest cross-frequency coupling to involve neural interactions between premotor areas and both insular and cingulate cortex as well as the medulla, which is precisely the pathway we propose to connect deep and cortical nodes within the RMBO network. A simple graph model of excitatory and inhibitory cells has been shown as proof of principle for cortical gamma modulation through respiration (modelled as sinusoidal input) [49]. The authors later concluded that respiration-locked brain activity has 2 driving sources [50]: On the one hand, respiration entrains OB activity via mechanoreceptors, as seen in local field potentials [25]. On the other hand, Heck and colleagues propose extrabulbar sources within the brain stem, which are of critical importance for the generation of the respiratory rhythm itself. Functionally, respiration thus appears to modulate higher oscillatory frequencies (e.g., gamma) for the purpose of integrating locally generated assemblies across the brain [51]. Our data now show that respiration–brain coupling (i) spans an even more extensive network including deep cerebellothalamic pathways; and (ii) involves a wider variety of oscillatory modulation than previously assumed. Importantly, our analyses demonstrate that the spectral specificity of respiration-related modulations within the RMBO network cannot be explained by mere changes in $CO_2$ alone (see S3 Text).

While the RMBO network presented here provides the most comprehensive account of human respiration–brain coupling to date, central research questions emerge as objectives for future work. First, having established the sources of respiration-related changes to neural oscillations, the transition from resting state to task context will illuminate the relevance of RMBOs for behaviour. Cognitive, perceptive, and motor performance have been shown to be modulated by respiration, warranting a closer assessment of the where (i.e., which site) and when (i.e., at which phase) of task-related RMBOs. Second, we have outlined functional pathways connecting the cerebellum to cerebral cortex via medullar and thalamic projections as well as the close link between OB and parahippocampal as well as prefrontal cortices. These putative hierarchies should be tested with directional measures of functional connectivity in order to reveal organisational relations within the RMBO network. Similarly, directed connectivity analysis can disambiguate bottom-up from top-down signals within the wider RMBO network and potentially illuminate the notably lateralised effects within it: Although lateralisation is not uncommon in well-established functional networks (e.g., related to attention; see [52]), it will be instructive for future work to validate whether RMBOs reliably prove stronger in one hemisphere (as was the case for insula and FEF) or whether there is a separate dynamic underlying the involvement of individual nodes.

In summary, our comprehensive investigation of respiration–brain coupling emphasises respiration as a powerful predictor for amplitude modulations of rhythmic brain activity across diverse brain networks. These modulations are mediated by cross-frequency coupling (linking respiratory to neural rhythms) and encompass all major frequency bands that are

thought to differentially support cognitive brain functions. Furthermore, respiration–brain coupling extends beyond the core RCN to well-known resting-state networks such as default mode and attention networks. Our findings therefore identify respiration–brain coupling as a pervasive phenomenon and underline the fact that body and brain functions are intimately linked and, together, shape cognition.

## Materials and methods

### Participants

A total of 28 volunteers (14 female, age 24.8 ± 2.87 years [mean ± SD]) participated in the study. All participants denied having any respiratory or neurological disease and gave written informed consent prior to all experimental procedures. The study was approved by the local ethics committee of the University of Muenster (approval ID 2018-068-f-S) and complied with the Declaration of Helsinki.

### Procedure

Participants were seated upright in a magnetically shielded room while we simultaneously recorded respiration and MEG data. MEG data were acquired using a 275 channel whole-head system (OMEGA 275, VSM Medtech, Vancouver, Canada) and continuously recorded at a sampling frequency of 600 Hz. During recording, participants were to keep their eyes on a fixation cross centred on a projector screen placed in front of them. To minimise head movement, the participant's head was stabilised with cotton pads inside the MEG helmet.

Data were acquired in 2 runs of 5-minute duration with an intermediate self-paced break. Participants were to breathe automatically through their nose while tidal volume was measured as thoracic circumference by means of a respiration belt transducer (BIOPAC Systems, Goleta, United States of America) placed around their chest. Continuous monitoring via video ensured participants were breathing through their nose instead of their mouth. Individual respiration time courses were visually inspected for irregular breathing patterns such as breath holds or unusual breathing frequencies, but no artefacts were detected.

For MEG source localisation, we obtained high-resolution structural magnetic resonance imaging (MRI) scans in a 3T Magnetom Prisma scanner (Siemens, Erlangen, Germany). Anatomical images were acquired using a standard Siemens 3D T1-weighted whole brain MPRAGE imaging sequence (1 × 1 × 1 mm voxel size, TR = 2,130 ms, TE = 3.51 ms, 256 × 256 mm field of view, 192 sagittal slices). MRI measurement was conducted in supine position to reduce head movements, and gadolinium markers were placed at the nasion as well as left and right distal outer ear canal positions for landmark-based co-registration of MEG and MRI coordinate systems. Data preprocessing was performed using Fieldtrip [53] running in MATLAB R2018b (The Mathworks, Natick, USA). Individual raw MEG data were visually inspected for jump artefacts and bad channels, but neither were detected. Both MEG and respiration data were resampled to 300 Hz prior to further analyses.

### MRI co-registration

Co-registration of structural MRIs to the MEG coordinate system was done individually by initial identification of 3 anatomical landmarks (nasion, left and right pre-auricular points) in the participant's MRI. Using the implemented segmentation algorithms in Fieldtrip and SPM12, individual head models were constructed from anatomical MRIs. A solution of the forward model was computed using the realistically shaped single-shell volume conductor

model [54] with a 5-mm grid defined in the Montreal Neurological Institute (MNI) template brain (MNI, Montreal, Canada) after linear transformation to the individual MRI.

## Computation of global field power

For the computation of global field power, the time courses of each channel of each participant were individually subjected to a continuous wavelet transform using a Morlet wavelet for 36 frequencies (from 2 Hz to 20 Hz in steps of 2 Hz and then in steps of 5 Hz up to 150 Hz). Next, we computed the absolute value of this complex-valued data and averaged these amplitude values across channels.

## Head movement correction

Based on previous respiration-related work from our lab [11], it was reasonable to assume that there would be respiration-induced changes in head position and/or rotation. Therefore, we computed individual Spearman correlations between the normalised respiration time course and head movement traces of translation and rotation (in x, y, and z direction, respectively) using the accurate online head movement tracking that is performed by our acquisition system during MEG recordings. Correlation coefficients were Fisher z-transformed and averaged across runs (for each participant) and across participants to yield group-level average correlation coefficients for all 6 head movement time courses. A series of $t$ tests revealed significant correlations between the respiration signal and translation in the x plane ($\rho$ [27] = −0.16, t [27] = −10.31, $p < 0.001$) as well as rotation in both x plane ($\rho$ [27] = −0.16, t [27] = −10.99, $p < 0.001$) and z plane ($\rho$ [27] = 0.16, t [27] = 11.08, $p < 0.001$; all $p$-values corrected for multiple comparisons using the Bonferroni–Holm method). S10 Fig shows head movement traces (translation and rotation) phase locked to respiration.

As some correlation between respiration and head movement was to be expected, it was critical to rule out that our results were confounded by these head movements. To this end, we used a correction method established by Stolk and colleagues [55]. This method again used the head movement tracking information (described above), i.e., 6 continuous signals (temporally aligned to the MEG signal) that represent the x, y, and z coordinates of the head centre ($H_x$, $H_y$, and $H_z$) and the 3 rotation angles ($H_\psi$, $H_\vartheta$, and $H_\varphi$) that together fully describe the head movement. We constructed a regression model comprising these 6 "raw" signals as well as their derivatives and, from these 12 signals, the first-, second-, and third-order nonlinear regressors to compute a total of 36 head movement-related regression weights (using a third-order polynomial fit to remove slow drifts). This regression analysis was performed on the power spectra of single-sensor (and single-voxel) time courses for analyses in sensor and source space, respectively, removing signal components that can be explained by translation or rotation of the head with respect to the MEG sensors.

In addition to controlling potential artefacts caused by head movement, we report a related control analysis for high-frequency muscle artefacts in S2 Text.

## Computation of MI and PTA

The MI quantifies cross-frequency coupling and specifically phase–amplitude coupling [24]. Here, it was used to study to what extent the amplitude of brain oscillations at different frequencies is modulated by the phase of respiration. To this end, the instantaneous phase of the respiration time course was computed with the Hilbert transform. Next, the time series at each sensor location were sequentially subjected to a continuous Morlet wavelet transformation at frequencies ranging from 2 to 150 Hz (with 2 Hz spacing below 20 Hz and 5 Hz spacing above 20 Hz) using the *cwtft* function in MATLAB with default settings. This function computes a

continuous Morlet wavelet transform using a Fourier transform-based algorithm. The Fourier transform of our wavelet is defined as

$$\Psi(f) = \pi^{-\frac{1}{4}} e^{-\frac{(f-f0)2}{2}} H(f), \tag{1}$$

where H(f) is the Heaviside function, and $f_0$ is the centre frequency in radians/sample. We then computed the amplitude envelope and smoothed it with a 300-ms moving average. MI computation was then based on the average amplitude at 20 different phases of the respiratory cycle. Any significant modulation (i.e., deviation from a uniform distribution) is quantified by the entropy of this distribution. To account for frequency-dependent biases, we followed previously validated approaches [9, 56] and computed 200 surrogate MIs using random shifts of respiratory phase time series with concatenation across the edges. The normalised MI was computed by subtracting, for each frequency, the mean of all surrogate MIs and dividing by their standard deviation leading to MI values in units of standard deviation of the surrogate distribution (see Fig 1B). Visual inspection confirmed that this removed the frequency bias in raw MI values (stronger MI for low frequencies compared to high frequencies). The computation resulted in normalised MI values for each channel, frequency, and participant. Following the established approach by Maris and colleagues [57], significance of these normalised MI values on the group level was determined by means of cluster-based permutation testing using *ft_freqstatistics* in Fieldtrip. This test controls for multiple testing and involves different steps. Specifically, we conducted a series of 1-tailed *t* tests of individual MI values at each frequency against the 95th percentile of the null distribution from the 200 surrogate MI values. t-Values were then thresholded at $p = 0.05$ and spectrally adjacent significant data points were defined as clusters. For each cluster, we then defined the cluster-level statistics as the sum of the t-values within every cluster. Each cluster was then tested for significance using Monte Carlo approximation. For this approximation, single subject MI spectra were randomly interchanged with the previously used 95th percentile spectra, and the *t* test was recomputed followed by clustering and computation of the cluster-level summed t-values. After repeating the randomisation procedure 5,000 times, the original cluster statistics were compared to the histogram of the randomised null statistics. Clusters in the original data were deemed significant when they yielded a larger test statistic than 95% of the randomised null data.

To assess oscillatory modulation over time, the PTA was computed from the smoothed, band-specific amplitude envelopes averaged across all sensors. Time points of peak inhalation were detected from the respiration phase angle time series using MATLAB's findpeaks function. For each time point of peak inhalation, global field power across all 36 frequencies was averaged within a time window of ± 1,000 samples centred around peak inhalation. The resulting 36 frequencies × 2,000 samples matrix was finally normalised across the time dimension, leading to z-scores of whole-head oscillatory power phase locked to the respiration signal. This analysis is equivalent to a wavelet-based time-frequency analysis. Computations were done separately for both MEG runs, normalised across the time domain, and finally averaged across runs and participants.

### Extraction of time series in source space

Source reconstruction was performed using the linearly constrained minimum variance (LCMV) beamformer approach [58], where the lambda regularisation parameter was set to 0%. This approach estimates a spatial filter for each location of the 5-mm grid along the direction yielding maximum power. A single broadband (2 to 150 Hz) LCMV beamformer was used to estimate the voxel-level activities across all frequencies.

## Rank optimisation and NMF

In our efforts to anatomically localise respiration phase–dependent modulation effects, we employed a spatially sparse variant of NMF to reduce the high (voxelwise) dimensionality in our data. Sparse NMF allows us to describe modulation indices across the brain as a low-dimensional combination of locally constrained network components, each of which provides a spectral profile for each participant. NMF has previously been applied for topological analyses of M/EEG data during tasks [59], at rest [60], and in decoding approaches [61]. As the MI is inherently nonnegative and the interpretation of NMF output matrices is rather straightforward, nonnegative factorisation in general was well suited to meet our demands. The sparse factorisation approach in particular has 2 key advantages over a regular NMF approach: First, regarding network identification, the sparsity constraints are highly beneficial in obtaining spatially specific rather than broad topologies, which was central for the next steps of our analyses. Second, these spatially specific topologies greatly enhance the precision with which time × frequency modulation characteristics can be displayed within one network component —the more distant voxels are included, the more component-specific modulations are diluted. In order to balance baseline differences between participants in preparation of the NMF, MI matrices of all 28 participants (20,173 voxels × 36 frequencies) were first normalised by their standard deviation [59]. These matrices were then averaged across both runs to yield one average matrix per participant. Individual matrices were transposed and concatenated to form one group-level input matrix (1,008 [frequencies × participants] × 20,137 voxels) for the NMF. To determine the number of main components to be extracted from NMF, we used the *choosingR* MATLAB function [62] that chooses the optimal rank based on singular value decomposition. Specifically, the function extracts the singular values of a data matrix (in our case, participants' normalised MI matrices; size 36 frequencies × 20,173 voxels) and computes the sum of all nonzero elements of its diagonal. The optimal rank is then determined as the number of singular values that accounts for 90% of all diagonal entries. Applying this procedure to participants' individual normalised MI matrices (36 frequencies × 20,173 voxels) returned a dimensionality of 18 as the optimal desired number of network components for the subsequent NMF analysis.

Subsequently, we initialised the *sparsenmfnnls* algorithm from the NMF toolbox for MATLAB [63]. The algorithm factorises the concatenated input matrix X as

$$X \approx AY, \tag{2}$$

with the nonnegative matrices A and Y aiming to minimise the following quantity:

$$X - AY_F^2 + \eta AY_F^2 + \lambda \sum\nolimits_{i=1}^{N} y_{i1}^2, \tag{3}$$

where η and λ are sparsity parameters. As NMF solutions vary as a function of their random starting position, we repeated this procedure 100 times and selected the best sparse solution based on its residuals. Two matrices were generated as the output of this procedure: First, the basis matrix A (1,008 [frequencies × participants] × 18 components) represents the participant-specific spectral profile, effectively quantifying each participant's relative contribution to the network components separately for each frequency. The basis matrix was reshaped to a 36 × 28 × 18 (frequencies × participants × network components) matrix for all further analyses. As the second NMF output, the coefficient matrix Y (18 components × 20,173 voxels) represents the spatial profile of the network components, quantifying each voxel's relative contribution to the components.

## Component-level statistical analyses

While most components represented a single focal location due to the sparsity constraints embedded in the NMF algorithm, 4 components comprised distinct subnetworks of 2 or 3 anatomical sites. Spatial maps of all 18 network components are shown in Fig 2. These maps were generated by thresholding full-brain maps (with a total of $n$ = 20,173 voxels) at the 99th percentile, yielding spatially specific maps with an extent of $n$ = 202 voxels. To determine the frequency range(s) for which the MI within a particular component was significant on the group level, we used the same cluster-based permutation approach described in detail in the section "Computation of MI and PTA." Here, this approach was used on all components together to correct for multiple comparisons across all 36 frequencies and 18 components.

## Modulation differences across NMF components

In order to compare modulation spectra across the RMBO network, we conducted a control analysis that compares frequency-specific effects across the 18 NMF components: For all components $1..n$ and each of the 36 frequencies used in our main analyses, we computed z-scores by comparing the MI value at frequency $i$ of a given component $j$ to the average MI value across the remaining 17 components:

$$z_{i,j} = \frac{\mu MI_{i,j} - \mu MI_{i,1..n \setminus j}}{\sigma MI_{i,1..n \setminus j}}.$$

This yielded a matrix of 18 components × 36 frequencies quantifying the difference between each component's MI spectrum relative to the grand average across all components. S10 Fig visualises this matrix thresholded at z = ± 2.33 (corresponding to $p$ = 0.01). A component's MI values were considered significantly different from the mean of all other components when the difference exceeded the critical z value. Components #3 and #14 (both located within the left cerebellum) showed greater than average modulation in the high gamma band, whereas components #6 (r. STG/r. temporal pole) and #11 (brain stem) were more strongly modulated in the delta and alpha band, respectively. Component #10 (bil. SMA) showed above average modulation in the beta and low gamma range. Finally, component #12 (bil. ACC) was less strongly modulated at low gamma frequencies than the grand average across components.

## LMEMs

We employed LMEM to investigate the relationship between the spatial organisation and spectral characteristics within the network of modulated components. LMEM models a response variable (in our case, modulation indices within a particular frequency band) as a linear combination of fixed effects shared across the population (i.e., anatomical coordinates of network components) and participant-specific random effects (i.e., modulatory variation between participants). To assess potential links between spatial and spectral component properties, we first computed each component's anatomical distance to the head centre as the vector norm of MNI coordinates in the x, y, and z plane:

$$r = \sqrt{x2 + y^2 + z^2}. \tag{4}$$

We then specified an LMEM to predict modulation indices of a particular frequency band within each component as a function of its distance to the head centre:

$$MI_j = \beta_0 + (\beta_1 + S_{1j}) * r + e_j. \tag{5}$$

For participant j, the MI is expressed as a combination of the intercept ($\beta_0$), the fixed effect of the component's distance to the head centre ($\beta_1$), and an error term ($e_j \sim N(0,\sigma^2)$). We accounted for between-participant variation by specifying a random slope ($S_{1j}$). An analogous approach was used to predict modulation indices within each component separately for each plane (see S1 Text and S1 Table).

## Hierarchical clustering

Having localised the sources of global field power modulations within a constrained subset of anatomical sites, our next aim was to characterise these sources in terms of their spectrotemporal fingerprints. This way, we hoped to reveal systematic patterns of phase-locked oscillatory modulations over time and/or frequencies within the cortical and subcortical network. To this end, we first computed the group-level average matrix of modulation indices for 20,173 voxels × 36 frequencies × 20 time bins. We used the anatomical distribution of each network component (thresholded at the 99th percentile) to reduce this matrix to a component-specific spatial map and aggregated 36 single frequencies into frequency bands as follows: delta (2 to 4 Hz), theta (4 to 8 Hz), alpha (8 to 2 Hz), beta (12 to 30 Hz), low gamma (30 to 70 Hz), and high gamma (70 to 150 Hz). This yielded one matrix (6 frequency bands × 20 time bins) per network component, all of which were concatenated to construct a distance matrix for the hierarchical clustering using the *hcluster* function within the Icasso toolbox for MATLAB [64]. This data-driven approach was employed to detect similarities of and differences between network components with regard to how oscillatory activity was modulated over the course of a respiration cycle. Following the suggested approach [64], we used visual inspection of the dendrogram to evaluate the clustering solutions. Based on a local maximum of the resulting silhouette value distribution, we settled on a total of 7 clusters (see Fig 3). We computed the average course of modulation indices over frequency bands within each cluster based on z-transformed spectral profiles of the contributing network components (as described above).

## Supporting information

**S1 Table. LMEM results for MI values as a function of x, y, and z planes.** Frequency bands were defined as described in the main text. Underlying data are provided in the folder "Supplementary Information" on the OSF directory. LMEM, linear mixed effect model; MI, modulation index.
(XLSX)

**S1 Fig. Range and distribution of sensor-level MI values.** Violin plot shows the distribution of individual MI values (depicted as dots) as well as the group-level median (white dot) across the whole frequency spectrum. Underlying data are provided in the folder "Supplementary Information" on the OSF directory. MI, modulation index.
(TIF)

**S2 Fig. Individual respiration traces and phase–amplitude spectrograms.** Left panels show single respiration traces centred around peak inspiration from each run. Right panel shows the individual phase–amplitude spectrogram averaged across all sensors. PTA values are shown as z-values, i.e., normalised within each frequency to reveal phase-related modulations. Underlying data are provided in the folder "Supplementary Information" on the OSF directory. PTA, phase-triggered average.
(TIF)

**S3 Fig. Individual spatial maps of cortical NMF components, projected on an inflated brain.** As in Fig 2, whole-brain NMF components were thresholded at the 99th percentile,

resulting in anatomical locations with an extent of $n$ = 202 voxels. Underlying data are provided in the folder "Supplementary Information" on the OSF directory. NMF, nonnegative matrix factorisation.
(TIF)

**S4 Fig. Individual spatial maps of subcortical NMF components, shown in frontal, sagittal, and transversal planes.** As in Fig 2, whole-brain NMF components were thresholded at the 99th percentile, resulting in anatomical locations with an extent of $n$ = 202 voxels. Crosshairs are positioned at the peak voxel of each location, and atlas labels are provided (corresponding to the nomenclature in Fig 2B). Underlying data are provided in the folder "Supplementary Information" on the OSF directory. NMF, nonnegative matrix factorisation.
(TIF)

**S5 Fig. Temporal modulation profiles of NMF components whose neural oscillations were significantly modulated by respiration.** **(A)** Cortical components plotted on an inflated brain surface. Polar plots show group-level normalised MI time courses averaged within frequency bands (delta to high gamma) over the entire respiration cycle. **(B)** Subcortical components plotted on transverse and sagittal slices of the MNI brain. Same format as A. Underlying data are provided in the folder "Supplementary Information" on the OSF directory. MI, modulation index; MNI, Montreal Neurological Institute; NMF, nonnegative matrix factorisation.
(TIF)

**S6 Fig. Comparison of frequency-specific modulations across components.** For each component and each frequency, we computed z-scores by comparing frequency-specific MI values of a particular component to the average across all other components (see main text for details). Opacity indicates significant differences (i.e., z = ± 2.33). Underlying data are provided in the folder "Supplementary Information" on the OSF directory. MI, modulation index.
(TIF)

**S7 Fig. Dendrogram of the hierarchical clustering performed on all 18 main components from the NMF analysis.** Dashed vertical line illustrates the cutoff criterion, yielding a total of 7 clusters. Cluster colouring is identical to Figs 3 and 4. Underlying data are provided in the folder "Supplementary Information" on the OSF directory. NMF, nonnegative matrix factorisation.
(TIF)

**S8 Fig. Top (left) and side view (right) of the RMBO network spanned by the 18 significant NMF components.** Numbering corresponds to Figs 2 and 4C as well as S3 and S4 Figs. Cluster colouring is identical to Figs 3 and 4 as well as S7 Fig. Underlying data are provided in the folder "Supplementary Information" on the OSF directory. NMF, nonnegative matrix factorisation; RMBO, respiration-modulated brain oscillation.
(TIF)

**S9 Fig. Waterfall plots show z-transformed amplitude modulation phase locked to the respiration cycle for the remaining frequency bands (from Fig 4D).** Clusters of NMF components are shown in the same order as in Fig 4C. Right panel bar graphs show the number of participants whose modulation within the respective component was strongest for the depicted frequency band (versus all other frequency bands). Coloured bars and circular segments mark NMF components for which the respective frequency band was significantly modulated by respiration phase. Underlying data are provided in the folder "Supplementary Information" on the OSF directory. NMF, nonnegative matrix factorisation.
(TIF)

**S10 Fig. Head movement across the respiratory cycle.** Top panel shows individual (grey lines) and group-level average time courses of the normalised respiration signal (bold). Bottom panel shows group-level average head movement signals phase locked to the respiration signal. Both translation (measured as Euclidean distance, yellow) and rotation (blue) are depicted as vector norms combining movement traces in x, y, and z directions. Underlying data are provided in the folder "Supplementary Information" on the OSF directory.
(TIF)

**S1 Text. Extended LMEM analysis of spatial patterns across planes.** LMEM, linear mixed effect model.
(DOCX)

**S2 Text. Control analysis for high-frequency muscle artefacts.**
(DOCX)

**S3 Text. Control analyses for spectrally unspecific modulations and movement-related changes in oscillatory power.**
(DOCX)

## Author Contributions

**Conceptualization:** Daniel S. Kluger, Joachim Gross.

**Data curation:** Daniel S. Kluger.

**Formal analysis:** Daniel S. Kluger, Joachim Gross.

**Funding acquisition:** Joachim Gross.

**Investigation:** Daniel S. Kluger.

**Methodology:** Daniel S. Kluger, Joachim Gross.

**Project administration:** Joachim Gross.

**Resources:** Joachim Gross.

**Software:** Daniel S. Kluger.

**Supervision:** Joachim Gross.

**Visualization:** Daniel S. Kluger.

**Writing – original draft:** Daniel S. Kluger, Joachim Gross.

**Writing – review & editing:** Daniel S. Kluger, Joachim Gross.

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
