## [Editor Report · Decision Letter 0]

2 Jun 2021

Dear Dr Kluger, 

Thank you for submitting your manuscript entitled "Respiration modulates oscillatory neural network activity at rest" for consideration as a Research Article by PLOS Biology.

Your manuscript has now been evaluated by the PLOS Biology editorial staff, as well as by an academic editor with relevant expertise, and I am writing to let you know that we would like to send your submission out for external peer review.

Please re-submit your manuscript within two working days, i.e. by Jun 04 2021 11:59PM.

Kind regards,

Gabriel Gasque

Senior Editor

PLOS Biology

ggasque@plos.org

---

## [Decision Letter · Decision Letter 1]

14 Jul 2021

Dear Dr Kluger,

Thank you very much for submitting your manuscript "Respiration modulates oscillatory neural network activity at rest" for consideration as a Research Article at PLOS Biology. Your manuscript has been evaluated by the PLOS Biology editors, by an Academic Editor with relevant expertise, and by four independent reviewers. You will note that reviewer 2, Jan Kujala, has revealed his identity.

The reviews of your manuscript are appended below. You will see that the reviewers find the work potentially interesting. However, based on their specific comments and following discussion with the Academic Editor, I regret that we cannot accept the current version of the manuscript for publication. We remain interested in your study and we would be willing to consider resubmission of a comprehensively revised version that thoroughly addresses all the reviewers' comments. We cannot make any decision about publication until we have seen the revised manuscript and your response to the reviewers' comments. Your revised manuscript would be sent for further evaluation by the reviewers.

As you will see from their detailed comments, the reviewers have some mixed views on the study, suggesting that the comprehensive nature of the work is nice, but they also raise various concerns related to potential confounds to your existing conclusions and a need to expand the work to provide more causal and mechanistic insights. In particular, reviewers 2 and 3 argue that more work is needed to rule out a confounding CO2-dependent mechanism, and reviewer 3 also asks that you provide additional controls to help rule out head movement confounds. While we appreciate that addressing these potential confounds would require a fair bit of additional work, such data seem necessary to strengthen the existing conclusions. In addition, to ensure the overall advance is sufficient for PLOS Biology, the revision should include additional analyses of the existing data as raised by reviewer 1 (point on coherence analyses) and reviewer 2 (points 2 and 3), and to provide some new data to address how behavior alters these findings (reviewer 4).

Having discussed the reviews with the Academic Editor, we think these should be addressed as the reviewers have requested, with additional data and analyses. Given the extent of the necessary revisions and their uncertain outcome, and that we would not be prepared to move forward with the work in their absence, we would also understand if you rather chose to seek publication elsewhere at this stage.

We appreciate that our requests represent a great deal of extra work, and we are willing to relax our standard revision time to allow you six months to revise your manuscript. 

**IMPORTANT - SUBMITTING YOUR REVISION**

*Resubmission Checklist*

*Published Peer Review*

*PLOS Data Policy*

*Blot and Gel Data Policy*

Sincerely,

Gabriel Gasque

Senior Editor

PLOS Biology

ggasque@plos.org

REVIEWS:

Reviewer #1: The manuscript by Kluger and Gross analyzes the effect of respiration on oscillatory brain activity in humans at rest. Using magnetencephalography in awake, resting subjects they provide evidence for widely distributed phase-amplitude coupling between respiration and different brain-intrinsic oscillations (delta, theta, alpha, beta, gamma). They further show that different frequency bands are differentially modulated at different locations, and that the identified networks have strong overlap with canonical networks, esp. the resting-state network.

The modulation of oscillatory brain activity by respiration has become a research focus during past years, resulting in a large body of literature from both, rodent and human studies. The observation that brain rhythms in humans are modulated by respiration is, hence, not entirely novel. However, the present article provides a very detailed, comprehensive picture of such modulations across brain regions and frequency bands. Moreover, the authors identified respiration-entrained sub-networks through clustering algorithms, and they map these networks onto known 'canonical' functional networks. The high resolution and degree of detail, the advanced methods of dimensionality reduction, the clustering and the mapping to the DNM are interesting and significant advances towards our previous knowledge.

As far as this reviewer can judge all methods have been applied very competently and methods and results are clearly laid out in the text and the illustrations. Nevertheless, there are some general concerns and several specific comments.

1. Nasal versus mouth breathing. Previous work by Tort and coworkers (mice, rats), Zelano and coworkers (humans) and others shows that, in most brain areas, respirational rhythms result from a feedback signal from nasal epithelia towards the brain. They are, hence, NOT caused by the rhythmogenic networks in the brainstem, but rather by sensory afferents (be it mechanosensitive or olfactory). It appears easy to test for the source of the described modulation by comparing the modulation of oscillations by breathing through the nose versus mouth (see, e.g., Zelano et al., 2016). I this reviewer's eyes, such a comparison would have introduced a clearly causal, rather than correlative finding into the paper. Without such (simple) manipulations, the work is mostly descriptive, though at a high level and, hence, still of great merit. 

2. Interpretation of data. Related to the first point, the introduction and discussion over-emphasize brain stem, cerebellar and other caudal sources of respiration, while more frontal areas are mostly considered as motor-related networks. Thus, the literature on feedback signaling of mechanical, thermal and also olfactory cues is not sufficiently represented and discussed. Some parts of the discussion have been written in very global terms, e.g. on the importance of rhythmic activity for information processing, active sensing etc. It would be good to relate the concrete findings of the present work to these concepts and focus even more on what has actually been shown.

3. Explanation of methodological limitations. In EEG and MEG recordings, contributions from deep subcortical networks can mostly be indirectly assessed by complex algorithms which leave some room for uncertainty and interpretation. Many readers might not be familiar with methods of source localization in EEG/MEG recordings. While the technical procedures have been well described in the Methods section it would be very helpful to give a brief, qualitative account of the underlying principles in the main text. This could, e.g. be placed in Results (around Figure 2) and/or Discussion and address the underlying principles as well as the power and limitations of such methods.

4. Analysis of inter-regional interactions. In order to test whether oscillations of a given frequency are synchronized between two regions, coherence is a well-established measure. This reviewer does not understand why this relatively simple method has not been applied to the oscillations, revealing a frequency-resolved measure of coordination across frequency and regions. Such data would also substantiate the notion of distinct sub-networks by using an independent approach. 

Specific comments (not ranked by importance):

5. Fig. 1C seems to show a modulation of low-frequency MEG signals by respiration (visible at the bottom right corner of the panel). For better visibility it would be helpful to show the PTA-phase-relation over TWO, rather than one, respiration cycle.

6. Upper graphs in in each panel of Fig. 2: the red horizontal line indicating significance threshold is barely visible in my version of the manuscript. 

7. For non-specialists, the large number of abbreviations makes the text difficult to read. Would it be possible to give full names upon the first mentioning of a technical / anatomical term within each section (i.e. for Introduction, Results, Discussion and Methods)?

8. The polar plots in Fig. 3 are not easy to read. As a minor correction, the authors might mark the 180° point as the point of inspiration (e.g., adding "I" or "In") and the 0° / 360° point as expiration. Moreover, it would probably be helpful to use a color code for the different frequencies, rather than grey scales.

Reviewer #2, Jan Kujala: The authors studied using MEG resting-state data how respiration influences neural signaling across the whole brain. The investigation was done via detailed spatio-temporo-spectral analyses. The results show that, globally, the whole spectrum of neural activity is affected by respiration. The results also show that different brain regions have distinct temporal and spectral profiles in how their activity is modulated as a function of the respiration phase. The results also suggest that these modulations have distinct spatial profiles where respiration is linked with low- and high frequency neural oscillations in lower and more lateral regions, respectively. The manuscript is well written and thorough and advanced analyses are conducted throughout the manuscript. However, I have some concerns whether the interpretations made based on the results are accurate. In particular, I find that additional analyses are required to demonstrate that the observed spatio-temporo-spectral specificity is meaningful instead of being an epiphenomenon of distinct profiles of neural activity and differential sensitivity of MEG in different brain regions.

Specific comments.

1. Globally, the results very clearly demonstrate that neural signaling across the whole investigated spectrum (2-150 Hz) is linked to respiration and the co-modulation is different across different frequencies. Moreover, the results show that these modulation spectra differ across brain regions. I am not, however, convinced that the present analyses demonstrate how meaningful the differences are and whether their relationship to oscillatory activity has a functional role. As the co-modulation of neural signaling with respiration seems to be very broad-band in nature as demonstrated by the authors as well as, e.g., Zeleno et al (cited by the authors), a strong possibility is that the observed temporo-spectral difference across brain regions are simply due to differences in the profiles of neural activity in them or due to how neural activity is detected with MEG. The same holds for the observed temporo-spectral specificity at the sensor-level. Namely, I find it possible that the majority of the findings within the manuscript could be explained, e.g., by a CO2-level dependent mechanism where all neural activity across the entire spectrum is influenced equally. If this holds, the observed differences could be due to the abovementioned distinct spectral profiles of neural activity across brain regions as well as due to differential delays in how CO2-levels follow the respiration phase in different brain regions. For the sensor-level results this explanation would mean that the spectral and temporal differences in the modulation index (Fig 1b) and phase-triggered average (Fig 1c) could represent the amount and/or SNR of neural activity across the entire brain and also the contribution of individual brain regions where a prominent relationship between neural activity and respiration can be detected due to these factors. Similarly, the voxel-level findings could be explained by variation in the detected spectral profiles of neural activity across brain regions as well as temporal differences in how the CO2-levels follow respiration. To dissociate between these possibilities (a more functionally meaningful role of the spatio-temporo-spectral specificity of the modulation vs. a simpler CO2-level dependent explanation), it would be valuable if the authors would, e.g., investigate the relationship between the modulation index and spectral profiles of neural activity both at the sensor-level and across different brain regions. A straightforward possibility would be to test how correlated the two measures are. If they would show only a weak link, it would suggest that there indeed is a functional role in the distinct spectral profiles of the co-modulation between respiration and neural activity, whereas a strong link between the measures would argue for a simpler explanation, e.g., a CO2-level dependent, global mechanism.

2. As regards the specificity of the findings across brain regions, I find that it would be beneficial if the analysis and illustration of non-negative matrix factorization based components would be expanded. Figure 2 demonstrates the spatial profiles of the components. However, the components are shown pooled together with lines indicating which cluster of voxels represents which component. It would be helpful if the component-specific spatial maps would be shown separately, either in the main manuscript or supplementary material. This would allow the readers to evaluate the whole set of voxels that primarily contribute to the component. Related to spatial profiles, it is unclear how the thresholding was done and what the results indicate. According to the manuscript the spatial maps were thresholded at the 99th percentile. Does this mean that they were thresholded at the 99th percentile across all components and voxels (18x20173)? That is, according the maximum value from amongst all voxels or components? Or was the thresholding component-specific? Or via some sort of a statistical evaluation as figure 2 states that "Colour bar indicates

corresponding p-values"? If the component-specific maps would be shown in the manuscript it would be possible to show the unthresholded maps so that it would be possible to evaluate how spatially specific each component is. Related to the spectral profiles of the components, it is unclear whether the modulation index values used in the statistical evaluation represent the relative average across all voxels or were modulation index values included only from the voxels that survived the 99th percentile thresholding. Regarding the spectral profiles of the components, it would also be beneficial to evaluate quantitatively how they differ between the components. At the moment, the manuscript presents for each component the frequencies where the modulation index is significant by comparing the actual data to surrogate data. In addition to this analysis, it could be evaluated, e.g., how individual components compare to the average of the other components. This could shed some light on whether the components are dissociated at specific, individual frequencies or whether the differences are more intricate with variation between component pairs in diminished and enhanced modulation index values across frequencies. 

3. Related to both above points, the authors applied linear mixed effect models (LMEM) to quantify whether the observed apparent differences in the cluster-level spectral profiles according to the anatomical location of the clusters (Fig. 3) is significant. The results showed that the distance to the head centre influenced gamma band modulation whereas regions low in the sagittal plane demonstrated stronger modulation than high regions. However, as these were the only LMEM analyses that were conducted, it is unclear how specific these relationships are. The spatio-specral relationship within the modulation index could be explored further by, first, evaluating for gamma-band the influence of the height of the sagittal plane and for low frequencies the effects of the distance to the center of the head and, second, by investigating relationship along the other planes as well and not restricting the analyses to the apparent differences. Moreover, if the modulation index values are strongly linked to the profile of neural activity (point 1 above), the LMEM analyses could be conducted also for the spectral profiles of neural activity in addition to the modulation index spectra.

Minor comments.

1. In the methods section (p22, line 535) it reads that "adjacent significant data points were defined as clusters". As the data within the manuscript are multidimensional (spatio-temporo-spectral) it would be helpful to clarify whether the adjacency means spatial, spectral or spatiospectral adjacency.

2. In the methods section (p22, line 557) it is stated that "The sensor covariance matrix used for

the LCMV-beamformer was computed across the whole data set.". I assume that this means that a single broad-band (2-150 Hz) beamformer was used to estimate the voxel-level activities across all frequencies. This could be clarified. 

Reviewer #3: The authors describe here the anatomical location and spectral profile of the modulation of neural activity along the respiratory cycle, from human MEG data during resting state. While the issue is important and topical, I was left wondering what are exactly the results and whether all necessary controls had been performed. As detailed below, there is little validation of the (complex and novel) methods used, little discussion of possible confounds, but also some discrepancies in how the data look like or how they are described. 

Major points

- The strength of MEG signals decreases with distance to sensors. Because the head moves with respiration, it follows that MEG signal power should vary across the respiratory cycle because of head movements. The authors provide a supplementary figure (supp Fig 7) illustrating the measured head movement with respiratory cycle and describe how they regressed out head motion (36 components) from the MEG data prior to data analysis. While this is certainly the best one can do with MEG data at the moment, whether this correction is sufficient or not is not known. 

- The results are described as follows in the Abstract: "high gamma modulation increased with distance to the head center, whereas delta and theta modulation decreased with height in the sagittal plane". I was left wondering whether such a physical, rather than physiological, description of the results corresponds to the fact that at least part of the results is due to the physical movement of the head with respiration, rather than to a true modulation of neural activity with respiration

- This impression was reinforced by the fact that the authors did not attempt to demonstrate that they precisely replicate some of the known modulations of neural activity by respiration measured in animals or with intracranial EEG in humans. For instance, they could have a region of interest analysis in the piriform cortex (see Zhou et al eLife 2019 for a parcellation of the human olfactory system) to verify that they reproduce known findings.

- The authors use a novel method, non-negative matrix factorization (NMF), which has been successfully used in other domains. However, this method has not been validated in the context of source localization of indices derived from MEG data. Could the authors provide some arguments / results to indicate that the results they present are not biased by the use NMF? 

- Finally, the authors relate their results to arterial CO2 fluctuations during respiration (discussion, lines 405-415). Could it be that arterial impedance changes across the respiration cycle, leading to changes in the magnetic field and hence to the measured MEG data? 

Other concerns

- Could the authors indicate inspiration and expiration on their graphs? Does phase 0 correspond to lungs full or empty? Does it depend on where the respiration belt is placed?

- There seems to be some discrepancy between the spectral profiles of MI (as shown in Fig2), that show peaks suggestive of frequency specificity, and the spectral profiles of component clusters, as shown in Fig3, that rather indicate a broad-band modulation. How come? For example, cluster C in Figure 3, composed of bilateral ACC and right FEF, has a spectral profile that looks quite different from the spectral profiles of bilateral ACC and right FEF as shown in Figure 2. 

- The description of the results is sometimes contradictory. Figure 2 shows modulation of neural activity in visual regions (bilateral cuneus, bilateral lingual sulcus), but the authors say with the exception of SMA, all cortical modulations sites are non-sensory, belonging to the default, attention or saliency networks (lines 255-258). 

Suggestions for improvement

- Fig 2, add some tick marks on the frequency axis, or write the value of the frequency for each vertical line

- phases are sometimes expressed in degrees (eg Figure 3), sometimes in radians (eg Figure 4)

Reviewer #4: The study by Kluger and Gross investigates the relationship between respiration and different frequencies of resting-state cortical activity in a sample of healthy adults. As with any research question that tackles the relationship between brain measures and a metric that is traditionally seen as an artifact, I had concerns regarding the confounds of the respiration itself and the MEG data. Nonetheless, the authors are transparent about this and do a decent job of removing any confounds that may contaminate the data, which is unsurprising given the authors' expertise in signal processing. However, there seems to be a disconnect between the rationale of the study and the approach, which limits my enthusiasm for the manuscript. My specific comments are below:

Introduction/Discussion:

My biggest concern with this paper is the disconnect between the rationale of the paper and the research approach. The authors do a nice job of laying out the previous work in this field, including in animals and using iEEG. However, the justification for the role of the cortex in oscillatory modulation of respiration (or vice-versa) is that automatic respiration can be overridden using top-down mechanisms. Nonetheless, in the actual experiment participants are instructed to rest with their eyes open, and respiration is recorded passively. Given that there is no overt top-down modulation of respiration in the experiment, a conclusion related to these mechanisms is impossible.

Methods/Results:

What was the procedure for rejection of artifactual data outside of respiration/movement?

In your figures, can you please put corrected p-values rather than percentile? Additionally, the data seems to be thresholded at p = .01 corrected via permutation testing, which is a relatively liberal threshold (and judging from the maps, may be too liberal given the additional clusters that are shown but are likely spurious given their topographic distribution. What does the data look like at a more reasonable p = .005 corrected? Did you use a cluster correction and if so, what was the k?

I am confused as to your rationale for both a whole-brain and ROI approach. Please clarify.

Is there any concern regarding the perceived "direcitonality" of the effects, whereby the frequency changes decrease or increase as a function of distance from the respiration?

---

## [Editor Report · Decision Letter 2]

12 Oct 2021

Dear Daniel,

Thank you for submitting your revised Research Article entitled "Respiration modulates oscillatory neural network activity at rest" for publication in PLOS Biology. I have now discussed your new version with the Academic Editor, and I am pleased to tell you that we will probably accept this manuscript for publication, provided you satisfactorily address the following data and other policy-related requests:

1) Blurb: Please provide a blurb which (if accepted) will be included in our weekly and monthly Electronic Table of Contents, sent out to readers of PLOS Biology, and may be used to promote your article in social media. The blurb should be about 30-40 words long and is subject to editorial changes. It should, without exaggeration, entice people to read your manuscript. It should not be redundant with the title and should not contain acronyms or abbreviations. For examples, view our author guidelines: https://journals.plos.org/plosbiology/s/revising-your-manuscript

2) Ethics: Please include ID number of your protocol approved by the local ethics committee of the University of Muenster.

3) Data: You may be aware of the PLOS Data Policy, which requires that all data be made available without restriction: http://journals.plos.org/plosbiology/s/data-availability. For more information, please also see this editorial: http://dx.doi.org/10.1371/journal.pbio.1001797

Note that we do not require all raw data. Rather, we ask for all individual quantitative observations that underlie the data summarized in the figures and results of your paper. For an example see here: http://www.plosbiology.org/article/info%3Adoi%2F10.1371%2Fjournal.pbio.1001908#s5

We note that you have stated that “The anonymised data supporting the findings of this study are openly available from the Open Science Framework (https://osf.io/6zbdu/).” However, the Data folder in the OSF link seems to be empty. Please double check and dismiss this request if I am wrong.

3.1) Please also include a README file that explains how the data was analyzed to generate the

the following figure panels: Figures 1bc, 2ab, 3AG, 4d, S1, S5, S6, S9, and S10.

3.2) Please also ensure that each figure legend in your manuscript includes information on where the underlying data can be found and that your supplemental data file/s has/have a legend.

We expect to receive your revised manuscript within two weeks. 

*Published Peer Review History*

*Early Version*

Sincerely,

Gabriel Gasque, Ph.D.,

Senior Editor,

ggasque@plos.org,

PLOS Biology

---

## [Editor Report · Decision Letter 3]

25 Oct 2021

Dear Dr Kluger,

On behalf of my colleagues and the Academic Editor, David Poeppel, I am pleased to say that we can in principle offer to publish your Research Article "Respiration modulates oscillatory neural network activity at rest" in PLOS Biology, provided you address any remaining formatting and reporting issues. These will be detailed in an email that will follow this letter and that you will usually receive within 2-3 business days, during which time no action is required from you. Please note that we will not be able to formally accept your manuscript and schedule it for publication until you have made the required changes.

PRESS

Sincerely, 

Gabriel Gasque, Ph.D. 

Senior Editor 

PLOS Biology

ggasque@plos.org